# How People in Eight European Countries Felt About the Safety, Effectiveness, and Necessity of COVID-19 Vaccination: A Cross-Sectional Survey

**DOI:** 10.3390/healthcare13030344

**Published:** 2025-02-06

**Authors:** Kristien Coteur, Marija Zafirovska, Aleksandar Zafirovski, Jelena Danilenko, Heidrun Lingner, Felix Bauch, Christine Brütting, Nicola Buono, Vanja Lazic, Liljana Ramasaco, Vija Silina, Lara-Marie Fuehner, Michael Harris

**Affiliations:** 1Academic Center for General Practice, Department of Public Health and Primary Care, KU Leuven, 3000 Leuven, Belgium; kristien.coteur@kuleuven.be; 2Medical Faculty, University of Ljubljana, 1000 Ljubljana, Slovenia; m.zafirovska27@gmail.com; 3Association of General Practice/Family Medicine of South-East Europe (AGP/FM SEE), 1000 Skopje, North Macedonia; 4General Hospital Jesenice, 4270 Jesenice, Slovenia; 5Department of Family Medicine, Riga Stradiņš University, LV-1007 Riga, Latvia; jelena.danilenko@gmail.com (J.D.); vija.silina@gmail.com (V.S.); 6MFD Health Group, LV-1055 Riga, Latvia; 7Hannover Medical School, Biomedical Research in End Stage and Obstructive Lung Disease Hannover (BREATH), German Center for Lung Research (DZL), 30625 Hannover, Germany; lingner.heidrun@mh-hannover.de; 8Hannover Medical School, Center for Public Health and Healthcare, Department of Medical Psychology OE5430, 30625 Hannover, Germany; fuehner.lara-marie@mh-hannover.de; 9Institute of General Practice and Family Medicine, Martin-Luther-University Halle-Wittenberg, 06112 Halle (Saale), Germany; felix.bauch@uk-halle.de (F.B.); christine.bruetting@uk-halle.de (C.B.); 10National Society of Medical Education in General Practice (SNaMID), 81100 Caserta, Italy; buono.nicola2@gmail.com; 11Association of Teachers in General Practice/Family Medicine, 10000 Zagreb, Croatia; vanjlaz@gmail.com; 12Nursing Department, Faculty of Technical Medical Sciences, Aleksandër Xhuvani University of Elbasan, 3001 Elbasan, Albania; lilianashyqyriu@hotmail.com; 13Department for Health, University of Bath, Bath BA2 7AY, UK; 14Institute of Primary Health Care Bern (BIHAM), University of Bern, 3012 Bern, Switzerland

**Keywords:** COVID-19, SARS-CoV-2, vaccine uptake, vaccine hesitancy, vaccination refusal, Europe, public health, population

## Abstract

**Background/objectives:** Attitudes towards COVID-19 vaccination vary globally, influenced by political and cultural factors. This research aimed to assess the views of people without a healthcare qualification in Europe on COVID-19 vaccination safety, effectiveness, and necessity as well as how well informed they felt. The secondary outcomes focused on how respondents’ views were affected by demographic and context factors and included a comparison by country of the level of feeling well informed. **Methods:** A mixed-method cross-sectional online survey in eight European countries, using convenience sampling. **Results**: A total of 1008 adults completed the survey, 60% of whom were female. While only 44.1% considered the vaccines safe, 43.5% effective, and 44.9% necessary, 80.0% had been vaccinated. Four in ten adults strongly agreed that they were well informed, while over a quarter did not answer the question. Younger respondents, well-informed individuals, and German respondents were more inclined to perceive COVID-19 vaccination as both effective and necessary. **Conclusions**: Motivations for vaccination included perceived health and social benefits, while concerns included a preference for “natural immunity”, the rapid development of the vaccine, and potential unknown long-term effects. A correlation existed between respondents feeling well informed about the different COVID-19 vaccines in their country and the likelihood of having been vaccinated.

## 1. Introduction

SARS-CoV-2 has led to over 777 million confirmed cases and more than 7 million deaths since the virus first emerged in December 2019 [1]. The virus transmits typically via respiratory droplets and leads to acute COVID-19 infection within 5 days. A severe infection with COVID-19 on the other hand, usually develops 7–10 days after symptom onset and could lead to hospitalisation, multiorgan disease, and death [2]. This is why many in research and development worked hard towards rapid development of effective COVID-19 vaccines. The global distribution and effectiveness of the vaccination during the pandemic represent a significant achievement in biomedical research. However, certain SARS-CoV-2 variants have shown increased resistance to these vaccines and treatments [3].

Approved vaccines undergo clinical trials to ensure safety and efficacy [4]. Subsequently, their effects are closely monitored, including for adverse drug reactions [5,6]. A consortium of researchers involved in a Cochrane network meta-analysis has continuously followed-up on the available evidence about the COVID-19 vaccines (covid-nma.com) [7]. By November 2021, it had been established that most available vaccines reduce or are likely to reduce the incidence of symptomatic COVID-19, and for some, there was good evidence that they reduce the severity of disease when infected [8]. There was little to no difference in serious adverse event rates between vaccines for COVID-19 and placebo in the general population [8].

Concerns regarding the safety and effectiveness of COVID-19 vaccines are commonly cited as reasons for vaccine hesitancy [9,10,11,12]. Media coverage of the side effects, along with shocking or controversial news stories, has further fuelled these concerns and generated negative or neutral sentiment [13,14]. People from different countries show different attitudes towards COVID-19 vaccination, suggesting that acceptance or rejection may be influenced by political or cultural opinions [15,16]. People in Western Europe were more willing to get vaccinated than those in Eastern Europe [17]. In African countries, one-third of adults were sceptical about vaccine efficacy and were not willing to get vaccinated [18]. In New Zealand, ethnic minorities and vulnerable communities tended to be distrustful towards their government, mostly due to their historical mistreatment [15]. In Asian countries, people showed strong trust in the government and were highly accepting (>80%) of vaccination [19]. There is evidence that when people distrust their government’s recommendations, they are more likely to trust friends, the internet, and books [20].

Ethical issues add to the discussions on the safety and effectiveness of COVID-19 vaccines, including whether the push for herd immunity prioritised profit rather than the safety of citizens; whether the rapid authorisation process compromised the need to ensure the vaccines’ quality, safety, and efficacy; and concerns about the lawfulness of achieving herd immunity through mandatory vaccination [21].

To better understand the varied factors influencing attitudes towards COVID-19 vaccination, we sought to investigate the perspectives of non-healthcare-trained individuals in eight European countries where there has been little relevant research, as perceptions on vaccine safety are known to vary considerably with nationality [22]. This study complements the existing literature, which often focuses on specific groups like immuno-compromised patients [23], childbearing women [24], parents [25], and healthcare workers [26] within a single country. Our goal was to assess the views of people without a healthcare qualification in Europe on the COVID-19 vaccination and to explore what influenced these views. We also aimed to compare how well-informed participants in participating countries felt about the different COVID-19 vaccines.

## 2. Methods

In the summer of 2022, we conducted a cross-sectional online survey study, as this is an effective way to assess patients’ attitudes and knowledge [27]. We used a mixed-methods approach to collect, analyse, and interpret the data [28], an approach reflecting the broad nature of the study’s aim [29]. The survey consisted of both closed and open-ended questions in order to achieve a more contextual understanding of the findings [27]. Partial responses were allowed.

### 2.1. Study Design and Participants

Individuals from eight European countries (Albania, Belgium, Croatia, Germany, Italy, Latvia, Macedonia, and Slovenia) were considered eligible if (a) they did not have any healthcare qualifications, (b) they were aged 18 years or older, and (c) they could complete the questionnaire in one of the languages in which it was hosted (Latvian, Russian, Macedonian, Slovenian, Dutch, French, Albanian, Croatian, German, Italian, and English). The study is reported following the STROBE guidelines [30].

### 2.2. Outcomes

The primary outcomes assessed the views of people without a healthcare qualification in Europe on (a) the COVID-19 vaccine’s safety, effectiveness, and necessity and (b) how well informed they felt about the different available vaccines.

The secondary outcomes were (a) how respondents’ views were affected by demographic and context factors and (b) a comparison by country of how well-informed respondents felt about the different available COVID-19 vaccines.

### 2.3. Questionnaire

The content of the questionnaire (English language version given in Appendix A) was informed by a review of the vaccine implementation literature [10,31,32,33,34,35,36,37], including surveys in Italy, France, Belgium, Canada, Slovenia, Japan, as well as a global survey on potential acceptance of COVID-19 vaccination [9,19,20,38,39]. The questionnaire in our study consisted of multiple-choice questions, 5-point Likert scales, and free-text questions regarding (1) sociodemographic data and an exploration of views and ideas on (2) vaccines, including willingness to be vaccinated; (3) vaccination status, including vaccine preference; (4) information status, requirements and sources; and (5) vaccination rates and campaigns.

The questionnaire was initially developed in English to allow input from all members of our multinational research team. We made slight adjustments to account for cultural differences between the countries, and it was validated at an international conference for general practitioners. Afterwards, the questionnaire was translated into each country’s main language(s), and these translations were validated through a back-translation [40]. To ensure accuracy and face validity, we piloted each translated version with at least two native speakers who had no prior medical or research experience. These pilots were conducted in the presence of a researcher, and the participants were asked to provide feedback on both the language and overall design of the questionnaire. Based on their suggestions, we made further small adjustments to improve the final version.

### 2.4. Recruitment

Given our objective to investigate a novel phenomenon [31], we considered convenience sampling to be appropriate [41]. Participants were recruited by the researchers using their personal and professional networks, with announcements in professional newsletters, blog posts, and a QR-code flyer that redirected possible participants to the survey. Based on an estimation of the total population across the participating countries, with 129 million adult inhabitants that do not have active healthcare [42], 385 responses were needed (confidence interval 95%). K.C. prepared this estimation with the SurveyMonkey sample size calculator and feedback from all co-authors about their national population statistics. We aimed to recruit at least 50 respondents from each of the eight participating countries.

### 2.5. Data Collection

Questionnaires were hosted online with Qualtrics software, Version May 2022 (Qualtrics, Provo, UT, USA). All data were collected anonymously, between 1 May 2022 and 30 September 2022. Intermittent team discussions of survey responses allowed us to continue data collection until data saturation for qualitative data was achieved.

### 2.6. Data Analysis

#### 2.6.1. Statistical Analysis

Likert scale responses were converted to numerical scores (“strongly disagree” = 1; “strongly agree” = 5). Descriptive statistics were used to report demographic data, whether respondents knew anyone who had been hospitalised for COVID-19, whether they had had all the regular vaccines on the vaccination schedule in their countries, whether they had been vaccinated for COVID-19, whether they felt well informed about the different COVID-19 vaccines used in their countries’ vaccination programmes, and their views on the vaccine’s safety and effectiveness and how necessary it was. We fitted an analysis of variance (ANOVA) model to investigate whether the between-country differences in the Likert scores for the statement “I am well informed about the different COVID-19 vaccines used in my country’s vaccination programme” were significant. We calculated Pearson’s correlation coefficient for correlations.

We used a binary logistic regression to test associations between participants’ views on the vaccine’s (a) safety, (b) effectiveness, and (c) necessity as well as demographic factors, whether they knew anyone who had been hospitalised for COVID-19, whether they had had all the regular vaccines on the vaccination schedule in their countries, whether they had been vaccinated for COVID-19, whether they reported being well informed about the different COVID-19 vaccines used in their countries’ vaccination programmes, the country that they lived in, the area that they lived in, and their employment status. In the regression analysis, “Slovenia”, “urban living”, and “currently employed” had the most participants in their groups and so were chosen as the reference variables for “country”, “area of living”, and “employment status”, respectively.

Tests were two-tailed, with statistical significance defined as *p* < 0.05. Data were analysed using IBM SPSS v28.

#### 2.6.2. Thematic Analysis

We analysed the qualitative data thematically. This method can be used across epistemologies and research questions [43]. It is an accessible form of analysis for researchers with little experience in qualitative methods [44]. M.H., K.C., H.L., N.B., and V.L., more experienced with thematic analysis, guided the other primary-care researchers in their analysis. Each country created a team of at least two researchers to independently code the free-text answers. These researchers were native speakers of the respondent’s language and processed the data inductively.

To faithfully represent the respondents’ views, their words were used to name the codes as much as possible. Constant comparison was used to identify patterns across the dataset. During this first stage of coding, the results were discussed in national teams, and disagreements were solved through discussion. Findings were discussed with the international team during intermittent project meetings. In the next stage, codes, categories, and themes were translated to English. We created a coding tree (Appendix A), which was validated through testing of additional surveys by each national team. Surveys were selected to maximise heterogeneity, specifically regarding age and sex of respondents. Disagreements were solved through discussion with all team members. After 334 sets of participant responses, we stopped data analysis because we found no new themes or interpretations of the data. Recurring themes were developed into descriptive accounts, as summarised below. These descriptions include deviant views to show that the themes represent recurring patterns rather than collective views and ideas. Examples of the findings are provided throughout the text with citations of single words or short statements.

The demographic data of respondents whose quotes are used are given in Appendix A.

### 2.7. Patient and Public Involvement

Both people with and without a healthcare qualification contributed to the questionnaire development during piloting.

### 2.8. Ethical Approval and Informed Consent

The study was approved by ethics committees in Belgium, Latvia, and Slovenia; in Macedonia, Italy, Albania, Germany, and Croatia, ethical approval was not required. Informed consent was obtained from all participants at the start of the survey.

## 3. Results

### 3.1. Participants

The team distributed the survey in eight European countries; 1008 respondents completed it. The largest demographic groups were female respondents (59.7%); those between 30 and 69 years old (67.6%); those with a high education level (72.5%); and those living in an inner-city environment (59.0%). Slovenia achieved the highest response rate (20.8%). Demographic details are given in Table 1.

### 3.2. Respondents’ Vaccination Rates

Regarding non-COVID-19 vaccinations, 925 (91.8%) respondents reported that they had completed the regular (non-COVID-19) vaccination schedule in their countries. In all, 806 (80.0%) had been vaccinated for COVID-19, and 650 (64.5%) respondents knew someone who had been hospitalised for COVID-19.

### 3.3. Views on Being Well-Informed

#### 3.3.1. Quantitative Analysis

Over 40% of respondents agreed or strongly agreed that they were well informed (Likert scores 4 or 5) about the different COVID-19 vaccines used in their countries’ vaccination programme (421; 41.7%) as opposed to over 20% who disagreed or strongly disagreed (Likert scores 2 or 1) (223; 22.1%) (Table 2). Over one-quarter of respondents did not answer this question (272; 27%).

Mean Likert scores ranged from 3.13 (SD: 1.46) in Albania to 3.79 (SD: 1.13) in Germany (Table 3). There was no significant difference in the responses by country (*p* = 0.48).

#### 3.3.2. Qualitative Analysis

Respondents’ information needs were related to knowledge regarding the effectiveness and safety of COVID-19 vaccination. They showed interest in how long the vaccine could protect them, its efficacy against long COVID, new variants of the virus, and its side effects: “*I’m too young to suffer from the side effects of the available vaccines, for a disease/virus which my body can fight on its own*” (R1).

Another recurring topic was the need for information about the composition of the vaccine and development and testing procedures. Respondents wanted an “*explanation about how it was done so rapidly*” (R2), more “*statistical data on effectiveness*” (R3), and study results that would contribute to (or prevent) an actual approval. Overall, they showed motivation to better understand the working mechanisms of COVID-19 vaccines.

However, some respondents also felt “*tired of COVID-19 information*” (R4) or competent enough to find answers themselves: “*I think that he who seeks, finds. There is no question at the moment that I have not had an answer to*” (R5).

### 3.4. Views on Safety, Effectiveness, and Necessity of COVID-19 Vaccinations

#### 3.4.1. Quantitative Analysis

Overall, 44.1%, 43.5%, and 44.9% agreed or strongly agreed with the statements “COVID-19 vaccines are safe”, “COVID-19 vaccines are effective”, and “COVID-19 vaccines are necessary”, respectively, more than those who disagreed or strongly disagreed (36.3%, 35.1%, and 37.0%; Table 4). The views for the “COVID-19 vaccines are necessary” statement were particularly polarised, with over half (50.9%) using the extremes of the scale and 7.0% reporting neutrality.

There was a strong correlation between the perceived safety, effectiveness, and necessity of COVID-19 vaccines. Respondents who perceived COVID-19 vaccines as safe also tended to consider them effective (r = 0.83, *p* < 0.001) and necessary (r = 0.82, *p* < 0.001). Those who perceived COVID-19 vaccines as effective also tended to perceive them as necessary (r = 0.83, *p* < 0.001).

Several factors were associated with increasing likelihood of the respondent agreeing that COVID-19 vaccines are safe, effective, or necessary (Appendix A). For each statement, there was a significant association with being vaccinated for COVID-19 and reporting being well informed about the different COVID-19 vaccines in their country (*p* < 0.001). For some statements, age and the country of residence were also significant: younger respondents were more likely to agree that COVID-19 vaccines are effective and necessary; respondents who chose “another country” were more likely to agree that COVID-19 vaccines were safe and effective; German respondents were more likely to agree that COVID-19 vaccines were effective and necessary.

#### 3.4.2. Qualitative Analysis

Complementing the quantitative findings, some respondents reported COVID-19 vaccines to be “safe enough” (R6). Some expressed their trust in science and regulatory affairs: “*Because I assume that the people who developed these vaccines know what they are doing. Studied for years for this and would not just put something on the market*” (R7). Others discussed personal experience, stating that they were “*without side effects after vaccination*” (R8) and that the vaccines are “*to protect myself and others around me*” (R9).

Many of the respondents who considered COVID-19 vaccination to be unsafe were worried about possible side effects: “*All vaccines, even if they have been the most important discovery of medicine in the last 300 years, could, however, in some very rare cases develop adverse reactions, even severe ones*” (R10). For some, it depended on the type of vaccine: “*Vector vaccines are not as safe as they should be*” (R11); “*mRNA vaccines: we are playing with fire despite the pseudo certainties of Pfizer*” (R12). Others were concerned about its long-term effects: “*Still a bit afraid of symptoms that would appear later on (higher risk for blood clots or heart attacks)*” (R13); “*even scientists cannot know if the vaccines will be completely safe in the long term*” (R14).

Regarding effectiveness, some respondents stated that the vaccine reduces disease severity. Reference was frequently made to protection from hospitalisation and death, possible prevention of long COVID, and prevention of the formation of new COVID-19 variations. Respondents viewed the evolution in incidence, hospitalisation, and mortality rates as an indication of the vaccines’ efficacy.

However, for others, effectiveness was related to the elderly population being vaccinated, e.g., “*The COVID-19 vaccines help older people get over the disease more easily*” (R15), or to the producer of the vaccine, e.g., “*some vaccines are more effective than others depending on who produces them*” (R16). Respondents reporting that vaccines were not effective described that it “*does not help against spreading the disease*” (R17) or was “*not effective in preventing the propagation of the virus*” (R18). They also expressed dissatisfaction because of negative personal experiences, such as “*still had COVID-19 after 2 doses*” (R19) and “*knew people with three doses admitted to serious intensive care*” (R20). They seemed to be worried because the vaccines “*do not seem effective for the current variants*” (R21), are “*not equally effective for everyone (variability)*” (R22), and “*people vaccinated with 2 doses were dying*” (R23). Some thought that the vaccines would “*compromises immunity*” (R24) or be harmful: “*they are effective for their purpose, but those effects are harmful for us*” (R25).

Expressions about necessity were often related to the impact of the COVID-19 pandemic on society and public health. On the one hand, respondents viewed the pandemic as a threat and felt an obligation to protect themselves and the people around them as well as a desire to end the pandemic and restrictive measures, such as lockdowns, as soon as possible. Some considered the impact on global health, citing that vaccines were “*in order to prevent infections from occurring in pockets of the unvaccinated population around the world that would lead to subsequent mutations (variants) of the virus with consequences that cannot yet be predicted by current scientific knowledge*” (R26), while others focused on their responsibility to protect vulnerable populations via “*herd immunity, particularly immunocompromised people who may not be vaccinated*” (R27). In contrast, some respondents believed that vaccination was only necessary for specific populations, such as the elderly or people with immune deficiencies: “*I believe that acquired immunity is better, but I do agree that vaccines are a far better option especially for immunocompromised people*” (R28).

Respondents who considered COVID-19 vaccination unnecessary stated a wide range of views. Some opinions were based on personal characteristics such as health status, e.g., “*I have never in my life been vaccinated and in very good health*” (R29), and age, e.g., “*I am young and healthy; chances are slim that I would be seriously ill from COVID-19. Therefore, I am not willing to take the risks that come with being vaccinated*” (R30). Respondents also referred to vaccination as experimentation with their health. For others, views about necessity were related to their views about the vaccines’ effectiveness, e.g., “*available vaccines have proven to be ineffective in the long term, and therefore unnecessary*” (R31), or about the vaccines’ safety, e.g., “*it’s less risky to get COVID than to get vaccinated*” (R32). Regarding the impact on society and public health, some respondents felt a sense of responsibility but did not believe vaccination was the answer. They believed that there were “*enough people with acquired immunity*” (R33) that “*protective measures are sufficient, if everyone sticks to them*” (R34), and that acceptance will develop over time: “*we will live with corona just as we live with the flu*” (R35).

## 4. Discussion

### 4.1. Principal Findings

This study explored views of a population, which was not trained in healthcare, across eight European countries. Based on 1008 completed questionnaires, 41.7% agreed that they were well informed about their vaccination possibilities compared with 22% who disagreed. Respondents wanted information about the safety, effectiveness, and the development of the vaccine. Although 35–37% of the respondents did not view COVID-19 vaccination as safe, effective, or necessary, 80% of all respondents reported that they had been vaccinated with at least one dose. Most participants expressed either a positive or negative view regarding the safety, effectiveness, and necessity of vaccination, with few (12%, 11%, and 7%, respectively) undecided. These opposing views were reflected in the qualitative data. Some respondents expressed trust in science and group immunity, while others felt that vaccination was redundant or only useful in vulnerable populations, such as the elderly or immunocompromised. Some reported conflicting views, doubting the vaccines’ effectiveness or having concerns about the long-term effect on their own health while also considering their responsibility towards society.

### 4.2. Comparison with Other Literature

While studies in the U.K., Portugal, the U.S., and Japan have shown vaccine hesitancy to be associated with younger age [45,46,47,48], younger respondents in the countries in our study expressed a higher likelihood of perceiving vaccination as necessary. The qualitative data suggest that this may have been driven by the desire to end lockdowns and return to a “normal life”. It could also relate to people not adhering to the restrictive measures or behavioural guidelines, which was described as a frustration by multiple respondents. Previous research showed non-linear and heterogeneous effects of age on the opinions of COVID-19 vaccines across European regions [49]. While older people generally consider the vaccine safe and effective and were more willing to get vaccinated against COVID-19 [9,50,51,52,53], a large cross-sectional study in Portugal observed higher hesitancy among participants aged 65 to 79 years compared to those aged 50 to 64 years [54].

Contrary to previous studies indicating higher levels of concern among women regarding COVID-19 vaccine safety and effectiveness [9,10,52,55,56,57,58], our data did not reveal a significant association between participants’ sex and their views on safety, effectiveness, and necessity. However, the country of residence significantly influenced respondents’ perspectives, possibly due to differing levels of confidence in vaccine safety and effectiveness, and these differences were also found between Eastern Europe and Southern or Western Europe [49]. Socioeconomic status may also play a role, as disadvantaged groups and religious individuals have shown higher polarization in their views on COVID-19 vaccines [52].

Although over one-third of respondents viewed the vaccine as unsafe, ineffective, and unnecessary, 80% of the sample had already been vaccinated against COVID-19 at least once. Since the onset of the pandemic, numerous SARS-CoV-2 variants have emerged, with the Omicron lineage surpassing earlier strains to become globally dominant [59]. The timing of the survey in the summer of 2022, when Omicron variants were prevalent in Europe, was a period marked by a resurgence in cases and deaths, with the case fatality rate increasing to 0.25% [60]. This may have contributed to the observed reduction in vaccine effectiveness against infection and transmission. However, real-world studies conducted during the Omicron era in high-risk populations (e.g., individuals aged ≥65 years, those with comorbidities, or immunocompromised individuals) showed that the updated COVID-19 vaccines had a reduced but still favourable effectiveness and safety in preventing severe outcomes [61].

Our study found a significant, positive association between being vaccinated for COVID-19 and reporting being well informed about the different COVID-19 vaccines. Increasing knowledge and understanding have been shown to reduce concerns about the vaccine side effects and rapid development and to increase acceptance [44]. The results indicate [62] a lack of information about safety, effectiveness, and vaccine development for the studied population.

### 4.3. Strengths and Weaknesses

This research in eight countries in the Western, Eastern, and Southern European regions allowed us to elicit a wide range of beliefs, values, and attitudes relating to COVID-19 vaccination. The combination of quantitative and qualitative questions yielded rich data concerning respondents’ perspectives on the safety, effectiveness, and necessity of COVID-19 vaccination. This approach provided insights into motivational factors, particularly those linked to the social context. Having access to a large and diverse participant pool from various countries and cultures makes this study more applicable and generalisable to a broad population. We used a process of back-translation to ensure that the different language versions of the questionnaire were equivalent to each other, and we piloted them carefully. The large sample gave robust statistical power and we achieved data saturation.

The content of the questionnaire was informed by previous studies on vaccination, but only face validity was obtained, so there is a risk that the measures were flawed and may not have truly captured the concepts that we intended to measure. Using convenience sampling created a risk of sampling bias and self-selection bias, as individuals needed to have technical skills to access and complete an online survey and access to a medium with internet connection. It may therefore be that our sample did not reflect the diversity of the target population, limiting the generalisability of our findings. Individuals with stronger opinions may have been more likely to participate, and this may have caused the polarisation of views that we found. We did not record socioeconomic status, which could also have been an influencing factor: those with lower socioeconomic status may have less access to accurate information through education, healthcare providers, and reliable media sources and less trust in institutions, and vaccination uptake is known to have differed by income level [63]. We gave the national study leads a choice of possible recruitment methods, and the methods used may have been different in each country. We do not know how many individuals received the survey invitation in each country, so we are unable to calculate response rates, and the number of respondents varied from country to country, limiting our ability to evaluate between-country difference in responses. For all countries, verification of inclusion and exclusion criteria was reliant on self-reporting, and individuals may have given incorrect answers so that they would be able to complete the questionnaire. The sample was sufficiently powered for an overall analysis but not for between-country analyses, so those findings should be interpreted with caution. While some respondents (*n* = 40; 4.0%) were from non-participating countries, this should increase the applicability of the findings to other populations.

### 4.4. Implications for Research and Practice

Research is needed to better understand the behaviour and attitudes of people that have a negative view about COVID-19 vaccines and yet agreed to be vaccinated. Longitudinal projects that study the potential changes in attitudes over time could also provide more insight in how to adequately inform the public. Participatory projects involving patients and the public in the development of educational interventions to increase knowledge regarding the vaccines’ testing and development, possible side effects, and the official recommendations could further increase uptake of COVID-19 vaccination. Finally, campaign developers could benefit from exploring decision-making frameworks, such as the Mindsponge theory [64], to effectively integrate the cultural influences that shape beliefs and attitudes.

## 5. Conclusions

This cross-sectional survey study explored Europeans’ views about the safety, effectiveness, and necessity of COVID-19 vaccination. Although 80% of respondents had been vaccinated, less than half agreed that COVID-19 vaccination was safe, effective, and necessary. Younger respondents were more likely to agree that COVID-19 vaccines are effective, and necessary. There was an association between respondents’ having been vaccinated against COVID-19 and their reporting that they were well informed about the different COVID-19 vaccines in their countries. In order to inform future public health educational interventions, research is needed to explore the drivers behind the attitudes and behaviour of vaccinated people with negative views about COVID-19 vaccines.

## Figures and Tables

**Table 1 healthcare-13-00344-t001:** Socio-demographics of the survey respondents (*N* = 1008).

Variable	Variable Category	Number (%)
Sex	Female	602 (59.7)
	Male	394 (39.1)
	Not given	12 (1.2)
Age	<30	292 (29.0)
	30–49	457 (45.3)
	50–69	225 (22.3)
	≥70	34 (3.4)
Country	Albania	132 (13.1)
	Belgium	105 (10.4)
	Croatia	141 (14.0)
	Germany	76 (7.5)
	Italy	90 (8.9)
	Latvia	92 (9.1)
	Macedonia	122 (12.1)
	Slovenia	210 (20.8)
	Another country	40 (4.0)
Highest education level	Elementary school	17 (1.7)
	High school	239 (23.7)
	Bachelor’s degree (or equivalent)	390 (38.7)
	Master’s degree (or equivalent)	274 (27.2)
	Doctoral degree	67 (6.6)
	Not given	21 (2.1)
Area of living	Inner city	595 (59.0)
	Suburban	222 (22.0)
	Rural	168 (16.7)
	Other	23 (2.3)
Lives alone	Yes	159 (15.8)
	No	849 (84.2)
Has children under 18 years old	Yes	315 (31.3)
	No	693 (68.8)
Employment status	Working	692 (68.7)
	Studying	120 (11.9)
	Unemployed	87 (8.6)
	Retired	69 (6.8)
	I prefer not to say	40 (4.0)

**Table 2 healthcare-13-00344-t002:** Likert scale responses and percentages for the statement “I am well informed about the different COVID-19 vaccines in my country’s vaccination programme”.

Likert Score	Number (%)
	*N* = 1008
1. Strongly disagree	103 (10.2)
2. Disagree	120 (11.9)
3. Neither agree nor disagree	92 (9.1)
4. Agree	201 (19.9)
5. Strongly agree	220 (21.8)
No response	272 (27.0)

**Table 3 healthcare-13-00344-t003:** Mean national Likert scale scores and standard deviations (SD) for the statement “I am well informed about the different COVID-19 vaccines in my country’s vaccination programme”.

Country	Mean Likert Score (SD)
	*N* = 1008
Albania	3.13 (1.46)
Belgium	3.41 (1.56)
Croatia	3.52 (1.35)
Germany	3.79 (1.13)
Italy	3.23 (1.32)
Latvia	3.32 (1.32)
Macedonia	3.27 (1.53)
Slovenia	3.51 (1.45)
Another country	3.97 (1.40)

**Table 4 healthcare-13-00344-t004:** Likert scale responses for the statements about safety, effectiveness, and necessity of COVID-19 vaccines (*N* = 1008).

Likert Score	COVID-19 Vaccines Are Necessary	COVID-19 Vaccines Are Effective	COVID-19 Vaccines Are Safe
	*n* (%)	*n* (%)	*n* (%)
1. Strongly disagree	230 (22.8)	181 (18.0)	206 (20.4)
2. Disagree	143 (14.2)	172 (17.1)	160 (15.9)
3. Neither agree nor disagree	71 (7.0)	113 (11.2)	125 (12.4)
4. Agree	169 (16.8)	254 (25.2)	219 (21.7)
5. Strongly agree	283 (28.1)	174 (17.3)	226 (22.4)
No response	112 (11.1)	114 (11.3)	72 (7.1)

## Data Availability

An anonymised version of the original dataset is available upon request.

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
