# Peer review of "How People in Eight European Countries Felt About the Safety, Effectiveness, and Necessity of COVID-19 Vaccination: A Cross-Sectional Survey"

_healthcare, 2025, doi:10.3390/healthcare13030344_

Round 1

Reviewer 1 Report (Previous Reviewer 1)

Comments and Suggestions for Authors

Dear Authors,

The document exhibits significant improvement compared to previous versions. However, a more comprehensive methodological description is needed. For example, it should specify whether the questionnaire allows for partial responses. Additionally, the tables should be self-explanatory; I recommend including notes to enhance their clarity.

Moreover, please address the ethical issues surrounding vaccination and hesitancy (DOI: 10.3390/vaccines10101602).

Finally, I suggest restructuring the discussion to cohesively integrate all sub-paragraphs rather than isolating a section solely for the study's limitations. A more detailed explanation of these limitations is warranted.

Best regards.

Author Response

Please find in the rebuttal attached below our answers.

Reviewer 2 Report (Previous Reviewer 3)

Comments and Suggestions for Authors

Thank you for addressing my comments/suggestions, the updated work is improved and I have no further comments on your work.

Author Response

We are pleased to hear that, thank you.

This manuscript is a resubmission of an earlier submission. The following is a list of the peer review reports and author responses from that submission.

Round 1

Reviewer 1 Report

Comments and Suggestions for Authors

Dear authors,

I read your interesting article, which addresses an original perspective on the COVID issue.

Below are some tips that I hope will be useful to you.

In the introducton section, too much emphasis is given to the mechanism of action and approval of the vaccine, which, although of primary importance, should be more concerned with the ethical aspects and hesitation towards vaccination (see: doi: 10.3390/vaccines10101602).

The gap intended to be filled, as are the aims, is very vague.

In the section relating to materials and methods, however, the sample size is mentioned without any reference to the software used or the statistician who handled these aspects, which you are kindly requested to indicate together with the report.

The methodology with which the questionnaire was administered is unclear. If this is a validated questionnaire, please include the questionnaire in all languages in the appendix to ensure the study's reproducibility.

The tables are not self-sufficient, but the results illustrated are very interesting; please improve.

The study's limitations are not mentioned. Please indicate in detail the numerous limitations of the study.

Please eview the abstract, discussions, and conclusions in light of the considerations.

Kind regards

Author Response

We have uploaded a pdf with our correspondence to Reviewer #1.

Reviewer 2 Report

Comments and Suggestions for Authors

Dear authors

This study is interesting in the aspect of peoples attitudes in the vaccination against a pandemic viral disease. The text is written well and minor corrections are required. 

Please include ongoing status and statistics of COVID-19 in the Europe. 

The methods and results are suitable. 

This study is valuable considering multi-national investigation of individuals (Albania, Belgium, Croatia, Germany, Italy, Latvia, Macedonia and Slovenia). 

Please also if possible cite to this paper: 

https://onlinelibrary.wiley.com/doi/abs/10.1002/jmv.27849

The reasons for vaccine hesitancy have been stated. 

If there are studies in 2024, please include in the manuscript. 

Comments on the Quality of English Language

Dear editor

The text is suitable.

Author Response

We have uploaded a pdf with our correspondence to Reviewer #2.

Reviewer 3 Report

Comments and Suggestions for Authors

Reviewer’s comments:

I am pleased to have a chance to review your paper titled “How people in eight European countries felt about the safety, effectiveness, and necessity of COVID-19 vaccination: a cross-sectional survey”. I think the paper topic is interesting and the work has a potential to contribute to the literature on public health and management, however, some parts of the work are still weak. Author(s) are required to take the following points to revise to improve the paper quality.

Specific comments:

1. Please strictly follow the format of the journal’s scientific article in the updated version (from numbering to labeling table, etc).

2. The paper lacks the literature review section, so please create this section and further elaborate on it in the revised version.

3. The discussion part is still weak due to the lack of clear justifications of the results. For example, author(s) are expected to provide detailed explanations/justifications of why so many respondents disagree the statement about safety, efficacy, and necessity of the COVID-19 vaccines, why younger people in the UK, Portugal, the USA, and Japan are likely to refuse to get vaccinated, or how goverment encourage younger people to participate in vaccination programs. In this sense, author(s) may consider using the thinking mechanism (mindsponge theory) and the core value-based decision making framework (mindspongeconomics) to back up your justifications and/or arguments about decision making process/mechanism of vaccination participation to enrich the content quality. It is noted that the core values include cost-benefit analysis, use values, social values, preference, etc. And actor(s) (e.g. young people) are more likely to engage in the vaccination if their core values are being met. Furthermore, author(s) may also consider using Culture Tower and the SM3D knowledge management theory to support your justifications about building vaccination culture associated with science/information-based solutions in the long run.

Comments on the Quality of English Language

NA

Author Response

We have uploaded a pdf with our correspondence to Reviewer #3.

Round 2

Reviewer 1 Report

Comments and Suggestions for Authors

Dear authors,

I want to express my regret if my previous suggestions were not well-received. However, I maintain the view that the paper, particularly considering the critical nature of the questionnaire upon which it is based, may need to meet the necessary scientific standards.

Kind regards